# Tourism Industry and Economic Growth Nexus in Beijing, China

**Yang Songling, Muhammad Ishtiaq ***  **and Bui Thi Thanh**

School of Management and Economics, Beijing University of Technology, Beijing 100081, China;
yang.sl@bjut.edu.cn (Y.S.); thanhkkt2009@gmail.com (B.T.T.)
* Correspondence: ishtiaqktk@163.com

**Abstract:** In the developing economy, tourism is the most visible and steadiest growing facade. Tourism is considered one of the rapidly increasing elements for economic development from the last two decades. Therefore, the proposed study used vector autoregression (VAR) model, error correction model (ECM), and the Granger causality to check the relationship between the tourism industry and economic growth based on the data of the Beijing municipal bureau of statistics from 1994 to 2015. Gross domestic product (GDP) is used as a replacement variable for the economic growth index, while internal tourism revenue is used as a tourism industry indicator. The study supports the tourism-led growth hypothesis proposed in the existing literature in a different survey of tourism and economic development. The results show that there is a strong relationship in the tourism industry and economic growth in the context of Beijing, and at the same time, tourism creates a more significant increase in long run local real economic accomplishments. The results of the VAR model confirm that in the long run, Beijing's economic growth is affected by domestic tourism, while the ECM model shows unidirectional results in the short term. Similarly, there is a one-way causal relationship between the tourism industry and economic growth in Beijing, China. The empirical results are in strong support of the concept that tourism causes growth.

**Keywords:** tourism; economic growth; cointegration test; VAR model; and ECM model

## 1. Introduction

Tourism-led growth literature proposed that the tourism industry plays a vital role in economic growth. In recent decades, tourism and the growth hypothesis has been a focal point for economists. The traditional analysis under debate is that development in tourism led to faster economic growth. Tourism is vital for developing countries as it contributes heavily to foreign exchange confronted by foreign exchange constraints (Oh 2005); furthermore, because tourism receipts together with export revenues well ameliorate current account deficits, promoting the tourism industry in those countries has become a primary development strategy.

Similarly, budget deficits may also be improved through increments of the tax revenues as tourism contributes to every single economic sector. On the basis of the assumption that tourism is an effective mechanism for economic growth, tourism promoters consistently lobby for investments and support in either way through infrastructure and developments (Becker and George 2011; Liasidou 2013) and the creation of new attractions (Getz 2008). Lew (2011) argued that tourism is the largest service sector in international trade and comprises a significant part of the global economy. According to united nation world tourism organization (2001UNWTO), tourism is within the top five bases of foreign export income in the world. Tourism contributes heavily to world revenue by providing an employment opportunity to the peoples of around 225 million and adds 6 billion dollars, which is 9% of total world revenue (Chou 2013). According to the World Travel & Tourism Council's annual research, and

Oxford Economics, Travel, & Tourism's contribution to world gross domestic product (GDP) outpaced the global economy for the sixth consecutive year in 2016, raised to a total of 10.2% to world GDP. The sector now supports 292 million people in employment—that is, 1 in 10 jobs all over. Travel & Tourism forecasts for the next ten years also show promising results with predicted growth rates of 3.9% annually in the same context.

Roe and Urquhart (2001) stated that in developing countries, tourism has become an important sector owing to its positive effects on infrastructure and superstructure, creating income, employment, and balance payment, as well as its high added value and impact on other industries. Tourism has been considered as a critical foundation of economic growth. Nowak et al. (2003) stated that to redouble tourism with economic growth, many governments carry out projects about infrastructure services. The importance and form of tourism have mostly changed especially, after 1990, by the effect of globalization. Binns and Nel (2002) highlight that tourism generally had sub-sectors. These sectors are motivated by input and output exchange, employment, transportation component, exporting, and other similar industries.

There is an interactive relationship between the tourism industry and economic growth. As the study is conducted worldwide by different researchers and over an unusual period, still there is a question mark as to whether the relationship between tourism and economic growth is unidirectional or bidirectional. Therefore, this sort of study is essential from time to time to check the results, implication, and reliability of the previous research. This study is conducted using the cointegration test, vector autoregression (VAR) model, error correction model (ECM), and Granger causality to analyze the relationship between the tourism industry and economic growth in Beijing, China for the period of 1994–2015. The study uses the three models together in order to gain a better understanding of the survey, which is missed in the limited empirical literature about tourism and economic growth. From a regional economic perspective, this study is first looking to the real link between the tourism industry and economic growth on one side and, at the same time, any importance from the economic growth toward the tourism industry in the context of Beijing, China. Second, the nexus between the tourism industry and economic growth will provide an experimental design with additional arguments to demonstrate the relationship between the tourism industry and economic growth. Third, the study is vital for a policymaker to show some discussions to make the right decisions for the future growth of Beijing's sustainable tourism sector. Although tourism is important and fast-growing, it is somewhat a facet of globalization, especially in an underdeveloped country. Excellent empirical literature shed light on the tourism industry pros and cons using cross country data. However, this paper uses the domestic data to unleash the determinant and consequence of tourism and economic growth.

Further, the proposed study is planned in the following order. Section 2 highlights the literature of relevant research, and domestic tourism development and economic growth. Section 3 presents the data and methodology added by the econometric model of the study. Chapter 4 describes the empirical results and discussion, aided by regression estimates of all the model used in the study, which are been explained in this section. Section 5 includes the conclusion, limitations, and some future research concepts. The references are listed under "References".

## 2. Literature Review

Economists emphasize the industrial properties of tourism growth in the economy while analyzing tourism. Schroenn and Tecle (2006) argued that the benefits of tourism are spread over a broader segment of society as compared with other sectors of the economy, because tourism is a multidisciplinary activity that involves several industries and draws upon a variety of skills. The revolutionary studies of the tourism led-growth hypothesis (Lea 1993; Sinclair 1998) unleashed that the possible results of the tourism industry are jobs creation, developed growth, and generate revenue for the government. On the basis of this hypothesis, Sinclair and Stabler (2002) stated that the potential strategic factor for economic growth is considered international tourism, as it provides the foreign exchange earnings, tourist spending, and an alternative form of exports.

Looking at the relationship between tourism and economic growth, the previous study shows ambiguous results using a different technique. There is no clear understanding of whether there are unidirectional or bidirectional results between the variables. Mohammadzade and Najafinasab (2009) used pool data and the Granger causality test to analyze the causal relationship between the tourism industry and gross domestic product in the selected Islamic countries during the period 1995–2005. The results claimed the existence of a one-way causal relation from gross domestic product towards a number of tourists.

Tang and Jang (2009) study the same sort of relationship while using Granger causality for U.S. economic growth and the results highlight that there is no unified relationship between the two variables. Similarly, Lean and Tang (2010) studied the same model for Malaysian market for the period from 1989 to 2010 and unleashed a bilateral relationship between tourism and economic growth only in short run, while in the long run, only economic growth affects tourism, which confirms the unidirectional link. Lee and Chang (2008) used a different test for organization of economic cooperation and development (OECD) and non-OECD countries to test the hypothesis, and the results demonstrate different outcomes for OECD and non-OECD countries, as OECD countries show a unidirectional causal relationship in the long run, while the non-OECD countries show a bidirectional relationship between tourism and economic growth.

Gunduz and Hatemi-J (2005) highlight that ambiguous results may stem from various reasons, such as using different econometric techniques; the relative weight of international tourism in the economies; missing crucial explanatory variables such as real exchange rates; and, most significantly, the poor quality of data in the empirical studies. The tourism-led growth hypothesis is strongly supported by Balaguer and Cantavella-Jordá (2002) in their research for Spain, which is the second largest recipient of international tourism revenues (5.9% of its GDP) next to the United States. Dritsakis (2004) found a bidirectional causal relationship between international tourism and economic growth in Greece, furthermore, both the hypothesis of tourism-led growth and growth-led tourism were valid for the Greek economy. Contrary to the Greek study, Sanchez Carrera et al. (2008) found a unidirectional causal relationship from international tourism toward economic growth for the Mexican economy and supported the tourism-led growth hypothesis. However, Oh (2005) results show no long-run cointegration connection for the Korea economy between tourism and growth.

Furthermore, it should be added that the tourism-led growth hypothesis for an economy highly depends on the relative weight in GDP from tourism. Kim and Chen (2006) results are in support of Dritsakis (2004), who find a bidirectional relationship between tourism and economic growth for Taiwan. That is, both the growth-led tourism and tourism-led growth hypotheses are valid to the economy of Taiwan, where the comparative weight of tourism revenues is similar to that of Korea. Proenca and Soukiazis (2005) tested the same sort of hypothesis for Portuguese regions and found that if the supply characteristics of the tourism sector are improved, it can be considered as a substitute method for attractive regional growth in Portugal. Cunado and De Gracia (2006) find some limited evidence for the African region.

*Domestic Tourism Development and Economic Growth*

Domestic tourism in China is as old as Chinese history. However, development in the tourism industry of China, Beijing is a new phenomenon, as China was closed to the outside world for an extended period. According to Sofield and Li (1998), domestic tourism in China dates back three thousand years ago, while international tourists have started coming into the country in the recent past. Therefore, it is essential to study the historical perspective of domestic tourism back to its emperors, scholars, and philosophers.

After the 1920s, modern domestic tourism started in China—first in Shanghai, where some travel agencies began working. These new travel agencies were used initially both for local tourism and outbound tourism (Qiao 1995), but this service of tourism was not the initiative for tourism in China, because those agencies were stopped their services forcefully by civil war and war against Japan.

Before the communist party ruling in the country, leisure travel for ordinary people was restricted only to the temple. This was mainly because of the lack of time, money, and facilities, as well as strict government control over internal migration (Gormsen 1995).

With time and especially after 1949, traveling and tourism was considered in the lifestyle of ordinary people and the government realized that they should always guard against (Zhang 1989). Still, however, there was some restriction and essential documents were needed, such as a permit to travel outside from one's local district. Over time, however, the tourism industry grew, and was considered one of the crucial elements of economic growth.

According to the facts and figures for tourism in China, in 2017, domestic tourism revenue was around CNY 4.57 trillion, with an increasing rate of 15.9%. The overall tourism industry contribution to the economy in every field—such as employment, where the direct employment is around 28.25 million people and indirect employment is 80 million peoples who contribute as a result to the total employment—is approximately 10.28%. Therefore, the study of domestic tourism and economic growth is vital to analyze the data in a different period. Thus, the hypothesis is developed like as follows:

**Hypothesis 1.** *There is no relationship between domestic tourism and economic growth in Beijing.*

**Hypothesis 2.** *There is a healthy interdependent relationship between domestic tourism and economic growth in Beijing.*

### 3. Data and Methodology

*3.1. Data EE*

The relationship between the tourism industry and economic growth is investigated in this study. Beijing Municipal Bureau of Statistics data are used from 1994 to 2015. The model is tested using E-views. There are numerous measures used in empirical studies as a proxy for the volume of the tourism industry (Gunduz and Hatemi-J 2005). Because literature does not specify any particular parameters for the tourism industry, this study used the natural log of tourism revenue (LNOR) (Gunduz and Hatemi-J 2005) as a proxy for the tourism industry and the natural log of gross domestic product (LNGDP) as an economic growth indicator. The reason for using the dataset in the natural log form is to avoid the problem of heteroscedasticity and make the result reliable and more evident.

*3.2. Econometric Model*

The proposed study used cointegration analysis, vector auto regression (VAR), Granger causality, and the error correction model (ECM) to analyze the role of tourism in the economic development of Beijing. The user-friendly VAR model is the most flexible, fruitful, and widely used model for the multivariate time-series analysis. It is used for the description and forecasting of active conduct of financial and economic time series. It provides a theory-based, real-time equation model, because it is the natural extension of univariate autoregression to multivariate time-series. The mathematical illustration of the VAR model used in this study is shown below.

$$X_t = \alpha + \alpha_1 X_{kt-1} + \text{€}_t \text{ where } \text{€}_t \sim n(0, \Omega),$$

where $X_t = (X_k)$ is a $(k \times 1)$ vector of time series variable; $\alpha$ is $(k \times 1)$ vector of intercept; $\alpha_1$ is $(1 \times k)$ coefficient matrices; and $\text{€}_t$ is $(k \times 1)$ unobservable, that is, a zero mean error term (white noise).

The estimation starts with unit root test to check the stationarity in the data. The augmented Dickey and Fuller (1981) unit root test is used to check the stationarity of domestic tourism revenue chain variable and GDP chain variable. The Dickey and Fuller (1981) ADF extension model is as follows:

$$X_t = \alpha + \alpha_1 X_{kt-1} + \text{€}_t \text{ where €t} \sim n\ (0,\ \acute{\Omega}),$$

where Xt = (Xk) is a (k × 1) vector of time series variable; $\alpha$ is (k × 1) vector of intercept; $\alpha$1 is (1 × k) coefficient matrices; and €t is (k × 1) unobservable, that is, a zero mean error term (white noise).

The estimation starts with a unit root test to check the stationarity in the data. The Dickey and Fuller (1981) unit root test is used to verify the stationarity of domestic tourism revenue chain variable and GDP chain variable. The Dickey and Fuller (1981) ADF extension model is as follows:

When Yt is a random step, there is no constant: $\Delta y_t = \beta y_{t-1} + \sum_{j=1}^{k} \varnothing_j \Delta y_{t-1} + \varepsilon_t.$

When Yt is a random step, there is constant: $\Delta y_t = \alpha_0 + \beta y_{t-1} + \sum_{j=1}^{k} \varnothing_j \Delta y_{t-1} + \varepsilon_t.$

When Yt is a random step with a constant around a random trend line:

$$\Delta y_t = \alpha_0 + \delta_t + \beta y_{t-1} + \sum_{j=1}^{k} \varnothing_j \Delta y_{t-1} + \varepsilon_t,$$

where $\Delta y_t = y_t - y_{t-1}$ is a series of time series under consideration; k is the latency length; $\varepsilon_t$ is white noise; the assay hypothesis is $H_0$: $\beta = 0$ (Yt is non-stop sequence of data); and $H_1$: $\beta < 0$ (Yt is the stop data sequence).

Similarly, the cointegration analysis is executed to check the integration for the variables. The Engle-Granger (Engel and Granger 1987) two-step cointegration and Johansen (1988) cointegration method is used to test the variables for integration. The test illustrates that if two series are individually integrated, but some linear combination of them has a lower order of integration, then the series is said to be cointegrated. The mathematical form of the cointegration model is shown below:

$$Xt = \beta + \beta 1Xt + \text{€}t.$$

Furthermore, the study checks the relationship in the long and short run using the following two cases:

Case 1, using the VAR model, no cointegration relationship is observed between the variables in the long run and short run.

Case 2, using vector error correction (VEC) model, there is a cointegration relationship between variables in the long run and short run.

## 4. Empirical Results and Discussion

### 4.1. ADF Unit Root Test

The estimation starts with the unit root test. A slow response of variables can be observed in the context of economics in the large data set. Therefore, linear regression in the econometric analysis leads to the problem of "pseudo regression". To avoid this problem, the stability of the time series should be checked before the model is established. By describing and analyzing the two sequences of LNGDP and LNOR, the sequence chart and difference sequence chart are shown in Figures 1 and 2.

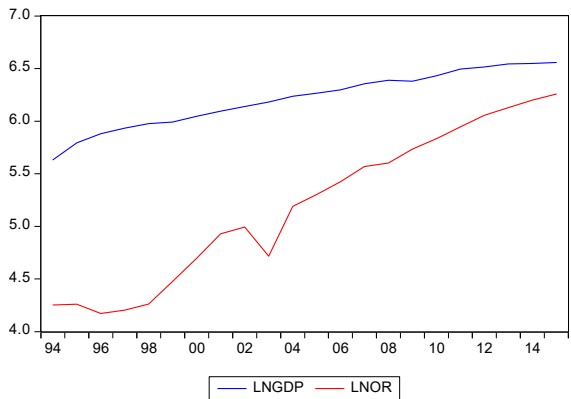

**Figure 1.** The trend graph of natural log of gross domestic product (LNGDP) and natural log of tourism revenue (LNOR) from 1994 to 2015.

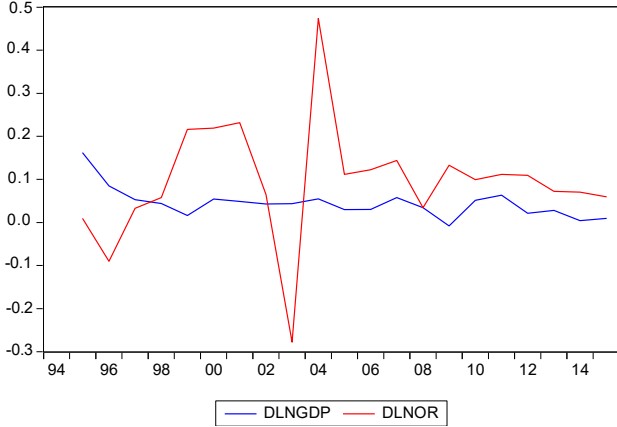

**Figure 2.** The trend graph of first-order difference LNGDP (DLNGDP) and first-order difference LNOR (DLNOR) from 1994 to 2015.

The chart in Figure 1 shows that both the variables increase simultaneously with time. Therefore, linear regression of these variables leads to the pseudo-problem. Stability of the time series is crucial in such a case. In the time-series data, there is always a chance of a stochastic trend, so it is essential to check the presence of unit root. The results of the augmented Dickey–Fuller (ADF) unit root test are shown in Table 1. The table shows that both the variables are non-stationary at a certain level. Both the variables are subject to the first-order difference, and it is shown in the table that they are stationary at the primary difference. Hence, at this level, the null hypothesis of unit root could no longer be accepted, which means that the series should be considered to be integrated at the first-order difference. Analyzing the results of ADF the study is necessary to proceed with cointegration and Granger causality between tourism and economic growth.

**Table 1.** Results of the augmented Dickey–Fuller (ADF) unit root test of variables. LNGDP—natural log of gross domestic product; LNOR—natural log of tourism revenue.

| Variables | Statistics of ADF | Marginal Value of 1% | Marginal Value of 5% | Marginal Value of 10% | Prob. | Durbin Watson (DW) | Conclusions |
|---|---|---|---|---|---|---|---|
| LNGDP | −3.276039 | −3.886751 | −3.052169 | −2.666593 | 0.0329 | 2.071201 | Non-stationary |
| LNOR | −0.132156 | −3.788030 | −3.012363 | −2.646119 | 0.9334 | 2.335525 | Non-stationary |
| DLNGDP | −5.155277 | −3.808546 | −3.020686 | −2.650413 | 0.0006 | 2.178049 | Stationary |
| DLNOR | −5.166582 | −3.808546 | −3.020686 | −2.650413 | 0.0005 | 2.039780 | Stationary |

Note: D represents the first-order difference to the time series.

### 4.2. Co-Integration Test

Generally, a cointegration test is used to investigate possible correlations among several time-series data in the long run. Sometimes, it has happened that no clear picture of the study comes through from only the cointegration analysis; therefore, to find more inside results and avoid the ambiguity of false regression, the current research will use the Engle–Granger two-step cointegration method developed by Engel and Granger (1987) and Johansen (1991) using the ordinary least squares (OLS). The results of the regression are shown in Table 2; the results demonstrate that both of the variables are significant.

**Table 2.** Co-integration test.

| Variable | Coefficient | Std. Error | t-Statistic | Prob. |
|---|---|---|---|---|
| C | 4.376291 | 0.115537 | 37.87775 | 0.0000 |
| LNOR | 0.353806 | 0.022049 | 16.04624 | 0.0000 |

$$\text{LNGDP} = 4.376291 + 0.353806 \times \text{LNOR} + \varepsilon_t.$$

Table 3 shows the results of the ADF residual test. According to the unit root test residual, the ADF of the residual unit root test is lower than the marginal value of 1%. Therefore, the residual sequence is stationary, indicating that the LNGDP and LNOR had a co-integration. Besides, to describe the relationship between the tourism industry and economic growth in Beijing, China, the determining lag length upon the VAR model is shown in Table 4.

**Table 3.** ADF test of residual.

| Variables | Statistics of ADF | Marginal Value of 1% | Marginal Value of 5% | Marginal Value of 10% | Prob. | DW | Conclusions |
|---|---|---|---|---|---|---|---|
| $\varepsilon_t$ | −5.361587 | −3.788030 | −3.012363 | −2.646119 | 0.0003 | 1.874306 | Stationary |

**Table 4.** Determining lag length upon the vector autoregression (VAR) model. LR—Likelihood; FPE—Final Prediction Error; AIC—Akaike Information Criterion; SC—Schwarz Information Criterion; HQ—Hannan–Quinn Information Criterion.

| Lag | LogL | LR | FPE | AIC | SC | HQ |
|---|---|---|---|---|---|---|
| 0 | 20.16988 | NA | 0.000354 | −2.271234 | −2.174661 | −2.266289 |
| 1 | 57.36864 | 60.44799 * | $5.63 \times 10^{-6}$ * | −6.421080 * | −6.131359 * | −6.406244 * |
| 2 | 58.89995 | 2.105552 | $7.93 \times 10^{-6}$ | −6.112494 | −5.629626 | −6.087767 |
| 3 | 63.21319 | 4.852396 | $8.29 \times 10^{-6}$ | −6.151649 | −5.475634 | −6.117031 |
| 4 | 68.17425 | 4.340924 | $8.71 \times 10^{-6}$ | −6.271781 | −5.402619 | −6.227273 |
| 5 | 71.06308 | 1.805523 | $1.39 \times 10^{-5}$ | −6.132886 | −5.070576 | −6.078487 |
| 6 | 76.26457 | 1.950556 | $2.32 \times 10^{-5}$ | −6.283071 | −5.027614 | −6.218781 |

* Indicates lag order selected by the criterion, LR sequential modified LR test statistic (each test at 5% level), FPE final prediction error, AIC Akaike information criterion, SC Schwarz information criterion, HQ Hannan- Quinn information criterion.

The first difference of the series, which is in stationary state, must have the proper lag length for future analysis. In Table 4, LR (Likelihood), FPE (Final Prediction Error), AIC (Akaike Information Criterion), SC (Schwarz Information Criterion), and HQ (Hannan–Quinn Information Criterion) were investigated to find the most proper lag length. According to this, we estimated 1 (as a value) as the most appropriate lag length. Therefore, the expected value "1" is used as the lag length in the analysis.

Tables 5 and 6 describe the results of Johansen cointegration analysis between LNGDP and LNOR. The results of the unrestricted cointegration rank test (trace and maximum eigenvalue) showed that there is one cointegration equation at a 5% significant level. That confirms that there is a long-term relationship between the tourism industry and economic growth.

**Table 5.** Unrestricted cointegration rank test (trace).

| Hypothesized | | Trace | 0.05 | |
|---|---|---|---|---|
| No. of CE(s) | Eigenvalue | Statistic | Critical Value | Prob.** |
| None * | 0.527936 | 18.43993 | 15.49471 | 0.0175 |
| At most 1 | 0.157479 | 3.427129 | 3.841466 | 0.0641 |

* denotes rejection of the hypothesis at the 0.05 level. ** MacKinnon et al. (1999) *p*-values.

**Table 6.** Unrestricted cointegration rank test (maximum eigenvalue).

| Hypothesized | | Max-Eigen | 0.05 | |
|---|---|---|---|---|
| No. of CE(s) | Eigenvalue | Statistic | Critical Value | Prob.** |
| None * | 0.527936 | 15.01280 | 14.26460 | 0.0380 |
| At most 1 | 0.157479 | 3.427129 | 3.841466 | 0.0641 |

Max-eigenvalue test indicates one cointegrating equation(s) at the 0.05 level. * denotes rejection of the hypothesis at the 0.05 level; ** MacKinnon et al. (1999) *p*-values.

*4.3. VAR Model*

4.3.1. Setup of VAR Model

It was estimated earlier that the variables are stationary at first order, thus the lag length value is "1" in this analysis. Furthermore, this study used the lagged one-time vector autoregression model by using Eview7. The estimated results of the model are shown in Table 7:

**Table 7.** Vector autoregression estimates.

| | LNGDP | LNOR |
|---|---|---|
| LNGDP(−1) | 0.678136 (0.05715) [11.8651] | 0.998738 (0.38319) [2.60636] |
| LNOR(−1) | 0.089291 (0.02128) [4.19663] | 0.636920 (0.14265) [4.46483] |
| C | 1.579499 (0.25101) [6.29264] | −4.226866 (1.68291) [−2.51165] |
| R-squared | 0.994529 | 0.972315 |
| Adj. R-squared | 0.993921 | 0.969239 |

LNGDP = 0.678136 × LNGDP(−1) + 0.089291 × LNOR(−1) + 1.579499; LNOR = 0.998738 × LNGDP(−1) + 0.636920 × LNOR(−1) − 4.226866.

The R-squared and Adj R-squared are higher, indicating that the equation has more top goodness of fit. The F-statistic is also higher, while the values of AIC and SC are smaller, which suggests that the model is generally significant. The established model can reflect a better relationship between LNGDP and LNOR.

Table 7 clearly shows that the influencing factors of LNGDP are it is lag factor LNGDP (−1) and domestic tourism revenue LNOR (−1), of which the main influencing factor is its lag factor LNGDP (−1), while local tourism revenue LNOR (−1) also affects it, but is not significant. The influencing factors of LNOR are its own lag factor LNGDP (−1) and domestic tourism revenue LNOR (−1).

4.3.2. Stability Test of VAR Model

The current study uses the AR root test to check whether the model is economically expressive and useful; the results of AR root test two series of joint stationarity with a lag length of 1 are shown

in Table 8, and the results show that the model is economically meaningful and stable. The value of the AR unit root test is 0.956866 less than 1, and all unit roots falling within the unit circle shown in Figure 3 confirm the basic rule of model stability. Therefore, it is derived from the results that there is a dynamic long-term equilibrium relationship between Beijing's tourism industry and economic growth. The established VAR model explains the characteristics of variables in the long-term.

**Table 8.** Roots of characteristic polynomial.

| Root | Modulus |
|---|---|
| 0.956866 | 0.956866 |
| 0.358191 | 0.358191 |

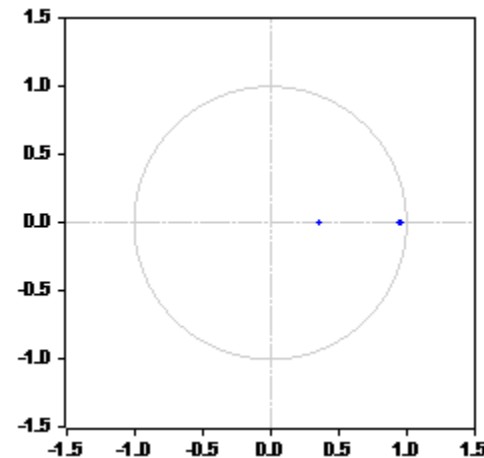

**Figure 3.** Inverse roots of autoregression (AR) characteristic polynomial.

### 4.3.3. Granger Causality Test

Bui (2018) used the causality test to check the long run exchange rate causal relationship for the Vietnam stock market. Granger causality is the most widely used model for causality between two variables. The cointegration analysis showed the long-term relationship between the tourism industry and economic growth in Beijing. Furthermore, this study conducted a Granger causality test with a lag length of 1 to confirm the causal relationship between the variables. The results of the Granger causality are shown in Table 9.

**Table 9.** Granger causality test results.

| Null Hypothesis: | Obs | F-Statistic | Prob. |
|---|---|---|---|
| LNOR does not Granger cause LNGDP | 21 | 17.6117 | 0.0005 |
| LNGDP does not Granger cause LNOR | | 6.79310 | 0.0179 |

The results indicate that both the variables are significant at a 5% level, and there is a causal relation between LNOR and LNGDP. Thus, it is stated that LNOR affects LNGDP in the case of Beijing, China, supporting the tourism-led growth hypothesis. From Table 9, it is clear that there is a causal relation between LNGDP and LNOR. Thus, it is also stated that Beijing's GDP also affects the tourism industry, supporting the growth-led tourism hypothesis and even the study (Dritsakis 2004). To sum up, the statement is that LNGDP and LNOR is the Granger reason for each other. Furthermore, it is stated that in Beijing, as a whole, the development of domestic tourism affects economic growth; economic growth also affects the development of local tourism identically.

### 4.3.4. Response and Variance Decomposition

In the practical application of the VAR model, it generally does not analyze how change in one variable affects another variable, but rather the dynamic influence of one error term of the model or the overall impact of receiving some kind of shock and dynamic structural analysis of variables. In addition, the economic interpretation of a single parameter is relatively difficult, where impulse response analysis is generally required. This article used the most common method of the Cholesky orthogonal impulse response, which is shown in Figures 4 and 5.

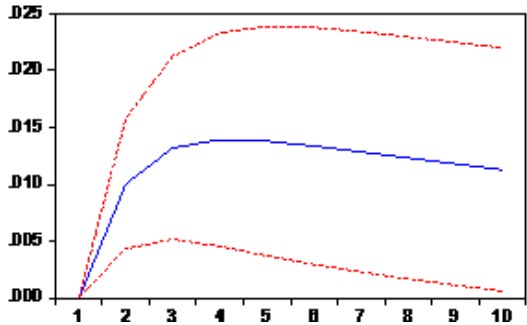

**Figure 4.** Response of LNGDP to LNOR.

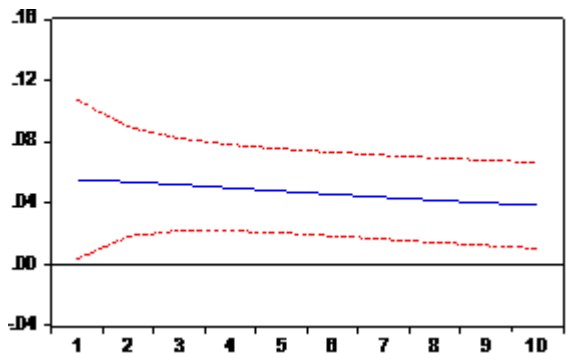

**Figure 5.** Response of LNOR to LNGDP.

The figure shows the dynamic variation between shocks after one standard deviation ($\pm 2$) of the random perturbation term. Figure 4 shows the change path of LNOR after being subjected to a standard deviation of LNGDP. The chart shows that economic growth has a substantial impact on Beijing's domestic tourism, which means that the change in GDP will directly affect Beijing's domestic tourism. An increase or decrease in GDP will lead to an increase or decrease in revenues from local tourism activities. Similarly, Figure 5 shows that the change in the path of LNGDP after being subjected to a standard deviation of LNOR, an increase or decrease in Beijing's domestic tourism income partly affects GDP. Overall, the GDP of Beijing is relatively stable; therefore, there is no considerable fluctuation while changing Beijing's domestic tourism. It started because of the reason that because of the stability in the economic conditions of Beijing, the impact of economic growth on the tourism industry is higher compared with the impact of the tourism industry on economic growth.

To move forward and discover more inside results and more confirmation the current study, a further variance decomposition analysis was conducted in the VAR model. Table 10 shows the results of the variance decomposition of LNGDP and LNOR.

**Table 10.** Variance decomposition of LNGDP and LNOR.

| Period | Variance Decomposition of LNGDP | | | Variance Decomposition of LNOR | | |
|---|---|---|---|---|---|---|
| | S.E. | LNGDP | LNOR | S.E. | LNGDP | LNOR |
| 1 | 0.018702 | 100.00000 | 0.00000 | 0.125392 | 19.39843 | 80.60157 |
| 2 | 0.027588 | 86.72390 | 13.27610 | 0.154157 | 25.03866 | 74.96134 |
| 3 | 0.034878 | 77.32994 | 22.67006 | 0.171932 | 29.23872 | 70.76128 |
| 4 | 0.040824 | 71.79544 | 28.20456 | 0.185497 | 32.32158 | 67.67842 |
| 5 | 0.045726 | 68.41139 | 31.58861 | 0.196726 | 34.61179 | 65.38821 |
| 6 | 0.049835 | 66.20935 | 33.79065 | 0.206354 | 36.34971 | 63.65029 |
| 7 | 0.053335 | 64.69148 | 35.30852 | 0.214752 | 37.69927 | 62.30073 |
| 8 | 0.056353 | 63.59378 | 36.40622 | 0.222151 | 38.77015 | 61.22985 |
| 9 | 0.058982 | 62.76867 | 37.23133 | 0.228710 | 39.63617 | 60.36383 |
| 10 | 0.061292 | 62.12900 | 37.87100 | 0.234554 | 40.34797 | 59.65203 |

From Table 10, when the forecast period is 1, the variance of economic growth changes from 100%, while the contribution from domestic tourism is 0%. We observe 19.39843% changes in domestic tourism from the level of economic growth data, which is 80.60157% of their participation. The analysis shows that as the forecast period increases, the gift of domestic tourism to self-variance gradually decreases, while the contribution of economic growth to the variance of domestic tourism slowly increases. By the tenth forecast period, the variation of the internal tourism data is 59.65203% from itself, 40.34797% from economic growth, 62.12900% from the financial growth variance from domestic tourism, and 37.87100% from itself.

Judging from the order of the variance decomposition analysis, the response of domestic tourism to economic growth is slightly higher than the reaction of economic growth to local tourism. This evidence shows that there is an information spillover effect between economic growth and domestic tourism—economic growth plays a guiding role.

*4.4. Vector Error Correction Model*

From unrestricted VAR models, the study reveals that there is a relationship between the tourism industry and economic growth in the long run, but the short-run relationship is not clear. To analyze the relationship between variables in the short run, the study used a vector error correction (VEC model). The VEC model is a restricted VAR designed for using non-stationary series that are known to be cointegrated. The VEC has cointegration relations built into the specification so that it restricts the long-term behavior of the endogenous variables to converge to their cointegrating relationships, while allowing for short-term adjustment dynamics. Using VEC models, the relationship between domestic tourism and economic growth of Beijing is shown in Table 11.

Table 11. Vector error correction estimates.

| Error Correction: | D(LNGDP) | D(LNOR) |
|---|---|---|
| CointEq1 | −0.233784 (0.13057) [−1.79044] | 2.178317 (0.79687) [2.73360] |
| D(LNGDP(-1)) | 0.115862 (0.16939) [0.68399] | 0.473391 (1.03377) [0.45793] |
| D(LNOR(-1)) | −0.033892 (0.03518) [−0.96328] | 0.019131 (0.21472) [0.08910] |
| C | 0.036149 (0.01057) [3.42071] | 0.076319 (0.06449) [1.18338] |
| R-squared | 0.363715 | 0.411513 |
| Adj. R-squared | 0.244411 | 0.301172 |
| Sum sq. resids | 0.006040 | 0.224962 |
| S.E. equation | 0.019430 | 0.118575 |
| F-statistic | 3.048650 | 3.729453 |
| Log likelihood | 52.67180 | 16.49677 |
| Akaike AIC | −4.867180 | −1.249677 |
| Schwarz SC | −4.668033 | −1.050531 |
| Mean dependent | 0.038155 | 0.099850 |
| S.D. dependent | 0.022352 | 0.141844 |
| Determinant resid covariance (dof adj.) | | $4.38 \times 10^{-6}$ |
| Determinant resid covariance | | $2.80 \times 10^{-6}$ |
| Log likelihood | | 71.08783 |
| Akaike information criterion | | −6.108783 |
| Schwarz criterion | | −5.610916 |

Note: CointEq1 is the estimated coefficient of error correction model (ECM) error term, the value in () is the standard error, the value in [] is the t statistic.

From the VEC model, the main influencing factors of the first order difference D (LNGDP) is D (LNGDP(−1)) and D (LNOR(−1)). The effect of D (LNGDP(−1)) is positive, and the impact of D (LNOR(−1)) is negative. The main influencing factors of the first order difference D (LNOR) are D (LNGDP(−1)) and D (LNOR(−1)), and the effect of D (LNGDP(−1)) and D (LNOR(−1)) is positive. The results also indicate that Beijing's tourism industry affects economic growth in the short-term, while economic growth also strongly influences the tourism industry, thus confirming the bidirectional relation between tourism and growth.

## 5. Conclusions

Beijing tourism industry has expanded rapidly over the past two decades and brought changes to China's various industries with strong functional characteristics. The tourism industry absorbs foreign currency with low cost, improves regional economic development, and increases economic vitality. Several industries such as telecommunication, transportation, insurance, finance, and commerce relating to tourism range from traditional to modern. Recently, a total of 5 million staff working directly in the Chinese tourism industry has been recorded. According to Zhu (2001), 25 million employees are working directly and indirectly in the Chinese tourism industry. This study conceptualized an econometric model of the tourism industry and economic growth for analysis. The results have shown a positive effect of tourism and economic growth for Beijing.

The tourism industry is the most important sector in Beijing, and it is stimulated by numerous policies set by China's government that encourage the growth of tourism industry, as well as the reformation of the People's Republic of China. Beijing's tourism has grown from a small number

of servicing units into the most important sector of the economy in Beijing, China. The Chinese government is aiming to double domestic spending on tourism soon. Therefore, this study is conducted to check the vital relationship between the tourism industry and economic growth with time-series analysis for the period of 1994 to 2015. The results of the cointegration analysis show that there exists a long-run cointegration relation between the total domestic tourism revenue and GDP for Beijing. Although the long run implication of the tourism industry and economic development is critical in the existing social sciences literature in a developing country, around the globe, government and international aid organizations are involved and show interest in the tourism industry, and invest a good amount of public funds to promote regional tourism. The results are consistent with the literature in the case of Beijing, where domestic tourism and economic growth are essential for one another in the long run.

The results of the VAR model indicate Beijing's economic growth will be affected by both the lag length domestic tourism revenue and the lag length economic growth, because tourism is becoming an important sector in the national economy and has been playing an active role in the economy of Beijing, China. Because of the rapid growth of domestic tourism, all related industries are growing. On the basis of the ECM model results in the short term, local tourism revenue and economic growth show a unidirectional relationship, supporting the findings in the previous study of Sanchez Carrera et al. (2008).

The proposed study results are significant for the policy maker about tourism in Beijing. This study shows robust empirical findings regarding the long- and short-run relationship between tourism and economic growth. The policy maker should give considerable attention to domestic and international tourism in Beijing. The sustainable position is also vital for the tourism industry in Beijing. The tourism industry can maintain its position and improve further by lowering the cost of traveling, living, and relaxation in the international tourism process. This study also used a different model at the same time to gain more insight information and fruitful results for future research.

*Limitation and Future Research Direction*

Although the study has significant implications and an essential contribution to the empirical literature in the context of tourism and economic growth, still it is not beyond the limitation. The results show a positive influence on tourism and economic growth in Beijing, but this does not mean that the development in Beijing is the result of the tourism industry as a whole. This is one of the most significant limitations of the study. This study used domestic tourism for Beijing as a tourism industry indicator. Future research can use international tourism together with domestic tourism to see the difference in linkage with economic growth. The study used data only for Beijing, so future research can use pool data from more cities in China to see overall results. More variables for the representation of the tourism industry, such as a number of tourists arriving, the number of nights stayed in the host country, and real exchange rate, among others, can also be used in future research with more advanced techniques. The prospective study can also control the crucial factors that influence the growth in Beijing's economy to find a clearer picture of the concept.

**Author Contributions:** Y.S. supervised the paper, M.I. write and organized the final draft and B.T.T. organize the data, all the author equally contributed.

**Funding:** There is no external funding for this paper.

**Conflicts of Interest:** The authors declare no conflict of interest.

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
