# Peer review of "Tourism Industry and Economic Growth Nexus in Beijing, China"

_economies, doi:10.3390/economies7010025_

Round 1
Reviewer 1 Report
Dear colleagues,
The article you present is of a good research quality and should find its way to readers. There are some misprints that you are advised to check and make alterations.
Page 4. domestic tourism in China is back from three thousand years where tourist from rural China, while international tourism is very recent in history
Page 7. In big data set some of the economic variables to become hysteresis.
Page 12. In the practical application of VAR model, generally it is not analyze how change in one variable affects another variable, but it analyze the dynamic influence of one error term of the mode
Page 14. the study reveals that the relationship between the tourism industry and economic growth in the long-term, but cannot be drawn conclusions about the relationship between the tourism industry and economic growth in the short-term.
Page 16. The tourism industry is one of the most important sectors of Beijing and it is stimulated by numerous policies set by China’s government which encourage the growth of tourism industry, as well as the reformation of People’s Republic of China, has resulted in a rapid development in this industry.
Page 16. that may useful for the economy of Beijing in long run.
Page16. organization are involved and showing interest in the tourism industry and by investing a good number of public fund to promote regional tourism.
Page 16. Because tourism becoming an important sector in the national economy and has been playing an active role in China, sometimes growing faster than the economy as a whole.
Page 16. all the related industries is booting up.
Page 16. Owing to domestic tourism development, and high demand it is needed that speeded up the development
Page 17. So it is concluded from the results that in short run domestic tourism and economic growth are undirected but still there is a strong connection between these two variables.
Page 17. The proposed study results is important for the current policy about the tourism in Beijing.
Page17. Although the industry is developed from last few year
Kind regards
Author Response
Page 4. Domestic tourism in China is back from three thousand years where tourist from rural China, while international tourism is very recent in history.
Response; Thank you very much for your useful suggestion and kind response. We have revised the particular section
Page 7. In big data set some of the economic variables to become hysteresis.
Response; Revised
Page 12. In the practical application of the VAR model, generally it is not analyze how change in one variable affects another variable, but it analyze the dynamic influence of one error term of the mode.
Response; Thank you again changes made in paper
Page 14. The study reveals that the relationship between the tourism industry and economic growth in the long-term, but cannot be drawn conclusions about the relationship between the tourism industry and economic growth in the short-term.
Response; incorporated in the particular section
Page 16. The tourism industry is one of the most important sectors of Beijing and it is stimulated by numerous policies set by China’s government which encourage the growth of tourism industry, as well as the reformation of People’s Republic of China, has resulted in a rapid development in this industry.
Response; Revised accordingly
Page 16. That may useful for the economy of Beijing in long run.
Response; Revised
Page16. Organization are involved and showing interest in the tourism industry and by investing a good number of public fund to promote regional tourism.
Response; Revised accordingly
Page 16. Because tourism becoming an important sector in the national economy and has been playing an active role in China, sometimes growing faster than the economy as a whole.
Response; Changes made
Page 16. All the related industries is booting up.
Response; Corrected
Page 16. Owing to domestic tourism development, and high demand it is needed that speeded up the development
Response; Revised accordingly
Page 17. So it is concluded from the results that in short run domestic tourism and economic growth are undirected but still there is a strong connection between these two variables.
Response; Revised
Page 17. The proposed study results is important for the current policy about the tourism in Beijing.
Response; Revised accordingly
Page17. Although the industry is developed from last few year.
Response; Thank you so much all the mistakes are incorporated as you suggested.

Reviewer 2 Report
The article "Tourism Industry and Economic Growth Nexus Beijing" is written in a fluent mode. However, I have some remarks to make:
The authors should further emphasize the importance of research and its limits.
The introduction must be developed and separated from the Literature Review.
Literature Review should be on subject and on methodology, on different levels to see the results of similar research done for other regions / countries.
The first two sentences in chapter 3 need to be reformulated. They have no substance, they are ambiguous and difficult to understand for the reader.
In chapter 3 there is no reference to bibliographic sources. The model is not designed by the authors. It is necessary to specify to whom the applied methods belong. At the Literature Review, the authors must point out who has used the methodology, in what context and with what results.
Chapter 4 is far too short. The authors must either develop it or integrate it into Chapter 3.
Conclusions should not be numbered.
The bibliography is relatively small. The last four bibliographic sources are numbered. The bibliography must be brought in a unitary form.
Author Response
The authors should further emphasize the importance of research and its limits.
Response; Thank you very much for your useful suggestion and kind response. We have revised the particular section
The introduction must be developed and separated from the Literature Review.
Response; Revised the introduction in the paper accordingly
Literature Review should be on subject and on methodology, on different levels to see the results of similar research done for other regions / countries.
Response; Revised the literature in the paper accordingly
The first two sentences in chapter 3 need to be reformulated. They have no substance, they are ambiguous and difficult to understand for the reader.
Response; changes are made accordingly
In chapter 3 there is no reference to bibliographic sources. The model is not designed by the authors. It is necessary to specify to whom the applied methods belong. At the Literature Review, the authors must point out who has used the methodology, in what context and with what results.
Response; thank you again good point changes are made accordingly.
Chapter 4 is far too short. The authors must either develop it or integrate it into Chapter 3.
Response; Revised
Conclusions should not be numbered.
Response; Revised
The bibliography is relatively small. The last four bibliographic sources are numbered. The bibliography must be brought in a unitary form.
Response; thank you so much the section is revised and more references are added accordingly.

Reviewer 3 Report
The paper undertakes an econometric analysis, searching for a relationship between increased tourism and economic growth, in Beijing, China.
I see two main problems with this paper:
1) It is very poorly written; not only the english language is clearly an obstacle for the authors, but also there are too many deficiencies and inaccuracies in the presentation of ideas, that make the paper unattractive to read (and even makes it difficult to understand the message the authors want to convey);
2) I have doubts on the meaningfulness of the analysis. Putting together data on tourism and growth of China's capital will naturally allow for a positive relationship (tourism brings revenues that might be used for investment); however, the main drivers of China's growth over the last decades are, obviously, others that the analysis overlooks.
Author Response
Reviewer 3 Comments:
1) It is very poorly written; not only the English language is clearly an obstacle for the authors, but also there are too many deficiencies and inaccuracies in the presentation of ideas, that make the paper unattractive to read (and even makes it difficult to understand the message the authors want to convey);
Response; Thank you very much for your useful suggestion and kind response. We have revised the articles and tried to improve the flow of paper to present a clear idea and thought for the reader.
2) I have doubts on the meaningfulness of the analysis. Putting together data on tourism and growth of China's capital will naturally allow for a positive relationship (tourism brings revenues that might be used for investment); however, the main drivers of China's growth over the last decades are, obviously, others that the analysis overlooks.
Response; Thanks again for the conceptual point, we revised the paper to improve the idea. This point also improves my understanding of nexus. Yes, it is true that tourism is not the only trigger for economic growth. But we just placed the tourism industries association in economic development for Beijing as the literature suggest. The tourism industry is not the only reason to change the whole scenario of Beijing development. Tourism industry is also linked with other industries like finance, insurance, telecommunication etc. yes the main them in the development of Beijing is other than tourism which is not subjected in this articles and maybe consider the limitation of the study, but the tourism industry is subjected to see the results and its contribution to GDP. We can take the example of a study in Turkey. The study of turkey show positive results though, turkey has other good policies and development program for the development of the economy. Similarly, Beijing is developed because of the other important policies of the government but the data on tourism and GDP show positive and significant results which confirm that tourism plays their role in economic growth and what literature suggests that there is an association between the tourism industry and economic growth for some region in the world economy.

Round 2
Reviewer 3 Report
I'm sorry, but I still have doubts on the scientific meaningfulness of the study, although some new arguments were adduced to the text, mainly in its final part. Furthermore, the english continues to be poor, with many grammatical errors in the text.